A longitudinal analysis of associations between traffic-related air pollution with asthma, allergies and sensitization in the GINIplus and LISAplus birth cohorts

Fuertes Elaine 1 2
Standl Marie 1 3
Cyrys Josef 4 5
Berdel Dietrich 6
von Berg Andrea 7
Bauer Carl-Peter 7
Krämer Ursula 8
Sugiri Dorothea 8
Lehmann Irina 9
Koletzko Sibylle 10
Carlsten Chris 2 11
Brauer Michael 2 11
Heinrich Joachim 1 heinrich@helmholtz-muenchen.de
1 Institute of Epidemiology I, Helmholtz Zentrum München, German Research Center for Environmental Health , Neuherberg , Germany
2 School of Population and Public Health, University of British Columbia , Vancouver , Canada
3 Institute for Medical Informatics, Biometry and Epidemiology, University of Munich , Munich , Germany
4 Institute of Epidemiology II, Helmholtz Zentrum München, German Research Center for Environmental Health , Neuherberg , Germany
5 Environmental Science Center, University of Augsburg , Augsburg , Germany
6 Department of Pediatrics, Marien-Hospital , Wesel , Germany
7 Department of Pediatrics, Technical University of Munich , Munich , Germany
8 IUF – Leibniz Research Institute for Environmental Medicine, University of Düsseldorf , Düsseldorf , Germany
9 UFZ Department of Environmental Immunology/Core Facility Studies, Helmholtz Centre for Environmental Research , Leipzig , Germany
10 Division of Paediatric Gastroenterology and Hepatology, Ludwig-Maximilians-University of Munich, Dr. von Hauner Children’s Hospital , Munich , Germany
11 Department of Medicine, University of British Columbia , Vancouver , Canada
Bousquet Jean
Electronic publication date: 2013 Nov 7
Publication date: 2013
Volume: 1
Electronic Location ID: e193
Received 2013 Jul 24; Accepted 2013 Oct 8
Copyright: © 2013 Fuertes et al.
Copyright year: 2013
Copyright holder: Fuertes et al.
License: This is an open access article distributed under the terms of the Creative Commons Attribution License, which permits unrestricted use, distribution, and reproduction in any medium, provided the original author and source are credited.
License URL: https://creativecommons.org/licenses/by/3.0/

Keywords: Asthma, Allergies, Air pollution, Birth cohort, Children, Long-term exposure, Traffic

Funding: Federal Ministry for Education, Science, Research and Technology Helmholtz Zentrum Munich Research Institute at Marien-Hospital, Wesel LMU Munich TU Munich IUF - Leibniz Research-Institute for Environmental Medicine at the University of Düsseldorf Federal Ministry for Environment (IUF Düsseldorf, FKZ 20462296) Helmholtz Centre for Environmental Research - UFZ, Leipzig Pediatric Practice, Bad Honnef European Community’s Seventh Framework Program (FP7-2007-2011) grant agreement number: 211250 Canadian Institutes of Health Research GINIplus study was mainly supported for the first three years of the Federal Ministry for Education, Science, Research and Technology (interventional arm) and Helmholtz Zentrum Munich (former GSF) (observational arm). The four, six, and 10 year follow-up examinations of the GINIplus study were covered from the respective budgets of the 5 study centres (Helmholtz Zentrum Munich (former GSF), Research Institute at Marien-Hospital, Wesel, LMU Munich, TU Munich and from six years onwards also from IUF - Leibniz Research-Institute for Environmental Medicine at the University of Düsseldorf) and a grant from the Federal Ministry for Environment (IUF Düsseldorf, FKZ 20462296). The LISAplus study was mainly supported by grants from the Federal Ministry for Education, Science, Research and Technology and in addition from Helmholtz Zentrum Munich (former GSF), Helmholtz Centre for Environmental Research - UFZ, Leipzig, Research Institute at Marien-Hospital, Wesel, Pediatric Practice, Bad Honnef for the first two years. The four, six, and 10 year follow-up examinations of the LISAplus study were covered from the respective budgets of the involved partners (Helmholtz Zentrum Munich (former GSF), Helmholtz Centre for Environmental Research - UFZ, Leipzig, Research Institute at Marien-Hospital, Wesel, Pediatric Practice, Bad Honnef, IUF – Leibniz-Research Institute for Environmental Medicine at the University of Düsseldorf) and in addition by a grant from the Federal Ministry for Environment (IUF Düsseldorf, FKZ 20462296). The exposure assessment research leading to these results has received funding from the European Community’s Seventh Framework Program (FP7-2007-2011) under grant agreement number: 211250. Elaine Fuertes was supported by the Canadian Institutes of Health Research (Sir Frederick Banting and Charles Best Canada Graduate Scholarship). The funders had no role in study design, data collection and analysis, decision to publish, or preparation of the manuscript.

==============================
Background. There is a need to study whether the adverse effects of traffic-related air pollution (TRAP) on childhood asthma and allergic diseases documented during early-life persist into later childhood. This longitudinal study examined whether TRAP is associated with the prevalence of asthma, allergic rhinitis and aeroallergen sensitization in two German cohorts followed from birth to 10 years.

Materials. Questionnaire-derived annual reports of doctor diagnosed asthma and allergic rhinitis, as well as eye and nose symptoms, were collected from 6,604 children. Aeroallergen sensitization was assessed for 3,655 children who provided blood samples. Associations between these health outcomes and nitrogen dioxide (NO2), particles with aerodynamic diameters less than 2.5 µg/m3 (PM2.5) mass, PM2.5 absorbance and ozone, individually estimated for each child at the birth, six and 10 year home addresses, were assessed using generalized estimation equations including adjustments for relevant covariates. Odds ratios [95% confidence intervals] per increase in interquartile range of pollutant are presented for the total population and per geographical area (GINI/LISA South, GINI/LISA North and LISA East, Germany).

Results. The risk estimates for the total population were generally null across outcomes and pollutants. The area-specific results were heterogeneous. In GINI/LISA North, all associations were null. In LISA East, associations with ozone were elevated for all outcomes, and those for allergic rhinitis and eyes and nose symptom prevalence reached statistical significance (1.30 [1.02, 1.64] and 1.35 [1.16, 1.59], respectively). For GINI/LISA South, two associations with aeroallergen sensitization were significant (0.84 [0.73, 0.97] for NO2 and 0.87 [0.78, 0.97] for PM2.5 absorbance), as well as the association between allergic rhinitis and PM2.5 absorbance (0.83 [0.72, 0.96]).

Conclusions. This study did not find consistent evidence that TRAP increases the prevalence of childhood asthma, allergic rhinitis or aeroallergen sensitization in later childhood using data from birth cohort participants followed for 10 years in three locations in Germany. Results were heterogeneous across the three areas investigated.

Introduction

The rapid rise in asthma and allergic diseases in recent decades suggests a role for environmental factors. Whether traffic-related air pollution (TRAP) contributes to the development of childhood asthma, allergy and related symptoms has been the topic of several studies. Although a review in 2010 concluded that the evidence of an association between TRAP and asthma onset and prevalence remains insufficient to infer a causal relationship (Tager et al., 2010), support for this association continues to build, including from genetic and gene-environment studies (Carlsten & Melén, 2012; Holloway et al., 2012). The evidence for other non-asthma allergic phenotypes, such as allergic rhinitis, eczema and aeroallergen sensitization, is generally weaker (Fuertes et al., 2013; Tager et al., 2010).

Several epidemiological studies have examined the link between air pollution and allergic diseases in children in early life (Clark et al., 2010; Morgenstern et al., 2007; Brauer et al., 2002; Gehring et al., 2002) and childhood (Carlsten et al., 2010; Morgenstern et al., 2008; Nordling et al., 2008; Brauer et al., 2007; McConnell et al., 2006). All of these aforementioned studies report positive associations with at least one respiratory or allergic health outcome, despite the challenges associated with accurately assessing air pollution exposure levels and allergic health status in young children. A study on older children (10–11 years) found no association between TRAP and general sensitization, although sensitization to house dust mites was associated with lifetime air pollution exposure (Oftedal et al., 2007). Two recent large cross-sectional multi-center European studies found no clear association between individually assigned TRAP exposures and asthma or wheeze at four to five and eight to 10 years in approximately 10,000 children (A Mölter, A Simpson, D Berdel, B Brunekreef, A Custovic, J Cyrys, J de Jongste, F de Vocht, E Fuertes , U Gehring, O Gruzieva, J Heinrich, G Hoek, B Hoffmann, C Klümper, M Korek, T Kulbusch, S Lindley, D Postma, C Tischer, A Wijga, G Pershagen, R Agius, unpublished data) and sensitization at four to six and eight to 10 years in approximately 7,000 children (Gruzieva et al., in press).

Only two longitudinal epidemiological analyses incorporating health data from birth up to eight years of age have been conducted. Gehring et al. (2010) reported positive associations between TRAP and the incidence and prevalence of asthma and the prevalence of asthma symptoms in a study of 3,863 Dutch children (Gehring et al., 2010). Positive associations were also found for allergic rhinitis, but only among non-movers. No associations were found for sensitization at eight years, despite a previous positive finding at four years in this same cohort (Brauer et al., 2007). Gruzieva et al. (2012) also found no association between TRAP and allergic sensitization at eight years in a Swedish birth cohort (Gruzieva et al., 2012), again despite a previous documented association at four years of age (Nordling et al., 2008). These results may suggest that the timing of air pollution exposure is important. This hypothesis, as well as the long-term impact of air pollution on allergic disease prevalence in later childhood, is best explored in the context of longitudinal birth cohorts (Bråbäck & Forsberg, 2009).

Recently, the 10 year follow-ups of the “German Infant study on the influence of Nutrition Intervention plus environmental and genetic influences on allergy development” (GINIplus) and “influence of Life style factors on the development of the Immune System and Allergies in East and West Germany plus the influence of traffic emissions and genetics” (LISAplus) birth cohort studies were completed. These studies are unique in their long-term and frequent follow-up, large sample sizes, extensive health, demographic and lifestyle information, as well as the availability of air pollution estimates at different time points during life (birth, six and 10 years). Previously published results from these cohorts suggest a possible adverse role of TRAP on symptoms before the first two years of life (Morgenstern et al., 2007; Gehring et al., 2002) and several allergic outcomes at six years (Morgenstern et al., 2008) for children living in the Munich (GINI/LISA South) area. In the northern part of these cohorts (Wesel area, GINI/LISA North), associations were found only with the prevalence of eczema (Krämer et al., 2009). The current study builds on these past efforts by examining whether nitrogen dioxide (NO2), particles with aerodynamic diameters less than 2.5 µg/m3 (PM2.5) mass, PM2.5 absorbance and ozone concentrations, the latter of which has not been previously considered in the context of the GINIplus and LISAplus studies, are associated with the prevalence of asthma, allergic rhinitis and aeroallergen sensitization among children followed for 10 years in three areas in Germany.

Methods

Study population

GINIplus is a prospective birth cohort of 5,991 children born at full-term and normal weight recruited in the areas of GINI/LISA South (a predominantly urban area) and GINI/LISA North (a predominantly rural area) between 1995 and 1998. Children with at least one atopic parent or sibling were allocated to an intervention study arm which investigated the effect of different hydrolyzed formulas consumed during the first year of life on allergy development (N = 2, 252) (von Berg et al., 2003). All children whose parents did not give consent for the randomized clinical trial or who did not have a family history of allergic diseases were allocated to the observation study arm (N = 3, 739). LISAplus is a population-based prospective birth cohort of 3,095 children born at full-term and normal weight recruited in GINI/LISA South, GINI/LISA North, Leipzig (LISA East; formally part of Eastern Germany) and Bad Honnef between 1997 and 1999 (original study size 3,097 but two participants withdrew their consent to participate). Detailed descriptions of the recruitment and follow-up strategy for both cohorts are available (Filipiak et al., 2007; Heinrich et al., 2002). Both studies were approved by the local Ethics Committees (the Bavarian Board of Physicians (reference numbers: 01212 and 07098), University of Leipzig (reference number: 345/2007), and Board of Physicians of North-Rhine-Westphalia (reference numbers: 2003355 and 2008153)) and written consent was obtained from all parents of the participants.

Questionnaire data

Demographic and health data were collected using parent-completed questionnaires administered when the child was one, two, three, four, six and 10 years old for GINIplus participants and six, 12 and 18 months and two, four, six and 10 years old for LISAplus participants. Only information collected from three years onwards is included in this study as it is difficult to accurately diagnose allergic health outcomes at very young ages.

A doctor diagnosis of asthma was defined as a positive response to “In the last 6/12 months, has your child been diagnosed with asthma?” A doctor diagnosis of allergic rhinitis was defined as a positive response to “In the last 6/12 months, has your child been diagnosed with hayfever or allergic rhinitis?” For GINIplus, this information was collected in one question at the three, four and six year follow-ups (diagnosis of hayfever or allergic rhinitis, as aforementioned). For the data collected at the 10 year follow-up of GINIplus and for all LISAplus follow-ups, this information was collected in two separate questions which were subsequently combined. If the follow-up covered a period of greater than one year, the prevalence of the diagnosis was asked separately for each year. Ultimately, yearly diagnoses for both asthma and allergic rhinitis were available for age three to 10 years. Eye and nose symptoms were assessed only at ages four, six and 10, and were defined based on two concomitant positive responses to “In the past 12 months, has your child had a clogged or itchy nose when he/she did not have a cold?” and “In the past 12 months, has your child had a clogged or itchy nose accompanied by watery eyes?”, which is consistent with the rhinoconjunctivitis definition used in the International Study of Asthma and Allergies in Childhood (Aït-Khaled et al., 2009).

Specific immunoglobulin E against common aeroallergens was assessed at ages six and 10 years using the standardized CAP-RAST FEIA method (ThermoFischer, Freiburg, Germany). Sensitization to inhalant allergens (SX1: Dermatoph. Pteroyssinus (house dust mites), cats, dogs, cladosporium (mold), birch, rye, mugwort and timothy grass) were measured by a screening test, followed by single specific allergen tests if the overall screening test was positive. The detection limit of the CAP-RAST FEIA method is 0.35 kU/L IgE; a test was defined as positive if the specific immunoglobulin E value was greater or equal to this limit. Aeroallergens were classified as either outdoor (birch, rye, mugwort and timothy grass) or indoor (house dust mites, cats, dogs, and molds).

Air pollution exposure

As part of the ESCAPE collaboration (www.escapeproject.eu), ambient concentrations of NO2, PM2.5 mass and PM2.5 absorbance were estimated for each child’s home address at birth, six and 10 years using land-use regression (LUR) models for children living in GINI/LISA South and GINI/LISA North (Beelen et al., 2013; Cyrys et al., 2012; Eeftens et al., 2012a; Eeftens et al., 2012b). For children living in LISA East, NO2 and PM2.5 mass estimates at the home address at birth and six years were derived from a similar LUR model developed as part of the earlier TRAPCA (Traffic Related Air Pollution and Childhood Asthma) collaboration which was conducted in the city of Munich, Germany (Cyrys et al., 2003; Brauer et al., 2003; Hoek et al., 2002). PM2.5 absorbance data are not available for LISA East. Ozone estimates from the “Air Pollution Modeling for Support to Policy on Health and Environmental Risk in Europe” (APMoSPHERE) project (www.apmosphere.org) were also available. For this latter pollutant, a 1 × 1 km resolution concentration map was developed across 15 European Union States using several European-wide datasets on monitored air pollution, land cover, altitude, transport networks, meteorology and population, as previously described (Beelen et al., 2009). Data from this map were used to assign ozone concentrations to the birth and six year home addresses of all participants. Table 1 summarizes the key characteristics of the models used to estimate individual-level air pollution concentrations in the current study. As no air pollution models were developed for the Bad Honnef area, children from this city were excluded from all analyses (N = 306).

Statistical analysis

All analyses were conducted using the statistical program R, version 2.13.1 (R Core Team, 2012). Differences in population characteristics were assessed using the Chi-square test. Longitudinal associations between air pollutants estimated at the birth addresses with the prevalence of doctor diagnosed asthma and allergic rhinitis, nose and eye symptoms and aeroallergen sensitization were analyzed using generalized estimation equations with a logit link (geeglm function from the geepack package (Haleko, Højsgaard & Yan, 2006)). An exchangeable correlation structure was used to account for repeated observations on the same individual.

Table 1 Characteristics of models used to estimate air pollution exposures.

Project	Areas	Air pollution sampling description	Pollutants	R 2	RMSE	Associated publications	
ESCAPE	GINI/LISA South	20 sites (PM)/40 (NO2) in Munich, Augsburg and small nearby towns sampled for three two-week intervals between 10.2008 and 11.2009	NO2	0.86	5.5	(Beelen et al., 2013; Cyrys et al., 2012)	
		PM2.5 mass	0.78	1.0	(Eeftens et al., 2012a; Eeftens et al., 2012b)	
		PM2.5 absorbance	0.91	0.2		
	GINI/LISA North	20 sites (PM)/40 (NO2) in Dortmund, Duisburg, Essen and smaller towns sampled for three two-week intervals between 10.2008 and 10.2009	NO2	0.89	4.3	(Beelen et al., 2013; Cyrys et al., 2012)	
		PM2.5 mass	0.88	0.9	(Eeftens et al., 2012a; Eeftens et al., 2012b)	
		PM2.5 absorbance	0.97	0.1		
TRAPCA	LISA East	30 sites sampled for four two-week intervals between 04.2004 and 03.2005	NO2	0.81	2.7	-	
		PM2.5 mass	0.55	0.1	-	
APMoSPHERE	GINI/LISA South
and North,
and LISA East	Air pollution data for 2001 obtained from Airbase (air quality database from routine air pollution monitoring covering 15 European member states; resolution 1 × 1 km)	Ozone	0.70	7.7	(Beelen et al., 2009)	
Notes.

R2 Model explained variance

RMSE Root-mean standard errors

All models were adjusted for sex, age, presence of older siblings (yes/no), parental history of atopy, parental education (originally defined using three categories based on the highest number of years of education of either parent, but collapsed into two categories due to low numbers in the lowest category: less than or equal to 10 years versus more than 10 years), maternal smoking during pregnancy, smoke exposure in the home (ever between birth and four years), contact with furry pets during the first year of life, use of gas stove for cooking during the first year of life, dampness or indoor molds in the home during the first year of life, intervention participation (GINIplus participants only), cohort and geographical area (for total models only). These confounders are the same as in previous analyses of these cohorts (Krämer et al., 2009; Morgenstern et al., 2008). Models for LISA East were not adjusted for intervention or cohort as only children for the LISAplus cohort were recruited from this area. Data on pneumonia infections in the first two years of life, which has been associated with air pollutants in a multi-centre study (MacIntyre et al., in press), and percent of total green space and population density in a 500 m buffer around the home address (GINI/LISA South and North only) were also available for sensitivity analyses. Elevated risks of disease were analyzed per interquartile range increase of each air pollutant. Odds ratios (OR) [95% confidence intervals (95 CI)] are presented for the total population and by geographical area.

As 59% of the study population reported moving at least once during the first 10 years of life, several sensitivity analyses were conducted to assess potential exposure misclassification. First, all models were rerun using air pollutants estimated at the six and 10 year home addresses rather than the birth address. Second, associations were assessed using the average of the birth, six and 10 year air pollution concentrations for NO2, PM2.5 mass and PM2.5 absorbance, and the average of the birth and six year concentrations for ozone (information at 10 years is not available). These averages should better reflect a true lifetime exposure for participants who moved between birth and 10 years. Lastly, associations were examined separately for children who had and had not moved during follow-up.

To address the potential importance of exposure timing, mutually adjusted models including both birth and six or 10 year address pollution concentrations were examined when the pollutants were not highly correlated (Pearson correlation coefficient <0.70). Each such correlation (for example, between NO2 assessed at the birth address and NO2 assessed at the six year address) was examined separately in the total and area-specific datasets.

Results

In total, 6,604 children had available information on at least one health outcome and air pollutant, 3655 of which also had available sensitization data at one time point (flow chart of study population provided in Fig. S1). Population characteristics and the prevalence of health outcomes at the age of 10 years are provided for the total population and by study area (Table 2), and for the subset of children who provided blood samples that were assessed for aeroallergen sensitization (Table S1).

Table 2 Characteristics of study participants.

	Total
N = 6604	GINI/LISA South
N = 3362	GINI/LISA North
N = 2551	LISA East
N = 691	
	n/N	%	n/N	%	n/N	%	n/N	%	
General characteristics									
Males	3386/6604	51.3	1747/3362	52.0	1304/2551	51.1	335/691	48.5	
Presence of older siblings	3021/6588	45.9	1398/3357	41.6	1378/2542	54.2	245/689	35.6	
Parental education									
Less than or equal to 10 years	2405/6573	36.6	759/3350	22.7	1342/2542	52.8	304/681	44.6	
More than 10 years	4168/6573	63.4	2591/3350	77.3	1200/2542	47.2	377/681	55.4	
Smoking									
During pregnancy	943/6248	14.7	430/3268	13.2	403/2494	16.2	110/666	16.5	
Ever in home (1–4 years)	2212/5663	39.1	887/2987	29.7	1096/2099	52.2	229/577	39.7	
Parental history of atopy	3750/6534	57.4	2232/3340	66.8	1221/2535	48.2	297/659	45.1	
Owned furry pet during early life	1126/6343	17.8	504/3237	15.6	444/2430	18.3	178/676	26.3	
Gas used in home during early life	469/6447	7.3	257/3301	7.8	106/2474	4.3	106/672	15.8	
Mold/dampness in home during early life	1330/5312	25.0	797/2889	27.6	379/1751	21.6	154/672	22.9	
Moved between one and 10 years	3197/5407	59.1	1739/2935	59.3	1061/1930	55.0	397/542	73.2	
Cohort									
GINIplus	4386/6604	66.4	2107/3362	62.7	2279/2551	89.3	0/691	0.0	
LISAplus	2218/6604	33.6	1255/3362	37.3	272/2551	10.7	691/691	100.0	
Intervention participationa	1935/6604	29.3	1031/3362	30.7	904/2551	35.4	0/691	0.0	
Health outcomes (at age 10 years)									
Doctor diagnosed asthma	164/4696	3.5	82/2589	3.2	68/1684	4.0	14/423	3.3	
Doctor diagnosed allergic rhinitis	460/4623	10.0	275/2528	10.9	140/1676	8.4	45/419	10.7	
Eyes and nose symptoms	628/4736	13.3	389/2585	15.0	173/1722	10.0	66/429	15.4	
Sensitized to aeroallergens	1100/2735	40.2	678/1581	42.9	303/867	34.9	119/287	41.5	
Sensitized to indoor aeroallergens	748/2732	27.4	449/1579	28.4	214/866	24.7	85/287	29.6	
Sensitized to outdoor aeroallergens	809/2734	29.6	509/1581	32.2	222/866	25.6	78/287	27.2	
Notes.

a Intervention only part of the GINIplus cohort.

Compared to the original cohorts, children included in this study and the subset who provided blood samples were less likely to have been exposed to furry pets early in life or to have moved at least once between birth and 10 years, but more likely to have participated in the nutritional intervention (GINIplus participants only) and to have a parent with more than 10 years of education and a history of atopy. Furthermore, the children included in the main analyses were less likely to have been exposed to smoke in utero. The subset of children who provided blood samples were less likely to have been exposed to tobacco smoke in utero and during early life (1–4 years) or later childhood (6–10 years).

Distribution of outcomes

The period prevalences of doctor diagnosed asthma and allergic rhinitis are presented in Fig. 1. In the total population, the annual prevalence of doctor diagnosed asthma and allergic rhinitis ranged from 1.1% to 3.5% and from 1.6% to 10.0%, respectively. The distribution of doctor diagnosed asthma and allergic rhinitis prevalence across areas was similar. Only at age nine and 10 were the rates of doctor diagnosed allergic rhinitis significantly different across areas. The pooled prevalence of reported eye and nose symptoms rose steadily with age and differed by area (3.7%, 6.8% and 13.3% for ages four, six and 10 years, respectively), as did the prevalence of aeroallergen sensitization (29.0% and 40.2% for ages six and 10 years, respectively). The prevalence of eye and nose symptoms and aeroallergen sensitization was highest in GINI/LISA South and lowest in GINI/LISA North at almost all ages.

Figure 1 Period prevalence of children with doctor diagnosed asthma (bars with diagonal lines) or allergic rhinitis (filled bars) at ages three to 10 years in the total population (A) and stratified by area: GINI/LISA South (B), GINI/LISA North (C) and LISA East (D).

Air pollution exposures

The distributions of annual average air pollution estimates are presented in Table 3 for the total study population and by area. At the birth addresses, mean NO2 concentrations were highest in GINI/LISA North (23.8 µg/m3), mean PM2.5 mass concentrations were highest in LISA East (17.5 µg/m3) and mean PM2.5 absorbance and ozone concentrations were highest in GINI/LISA South (1.7 × 10−5/m and 45.8 µg/m3, respectively). As the GINI/LISA North area is affected by the neighboring industrial Ruhr area in Germany, the elevated levels of PM2.5 mass but not PM2.5 absorbance in this area may suggest that there are PM2.5 mass sources other than traffic. The correlations between NO2 and the other pollutants were moderate in the pooled data (0.30, 0.32 and −0.50 for PM2.5 mass, PM2.5 absorbance and ozone, respectively).

Table 3 Distribution of estimated annual average concentrations of NO2, PM2.5 mass, PM2.5 absorbance and ozone at the birth addresses in the total dataset and per area.

Air pollutant	N	Min	0.25	Median	Mean	0.75	Max	IQR	
NO2 (µg/m3)									
Total population	6485	11.5	18.9	22.2	22.4	25.0	62.8	6.1	
GINI/LISA South	3306	11.5	17.3	20.7	21.7	25.4	61.1	8.1	
GINI/LISA North	2491	19.7	21.8	23.2	23.8	25.1	62.8	3.3	
LISA East	688	18.5	18.7	18.8	20.8	22.4	34.8	3.7	
PM2.5 mass (µg/m3)									
Total population	6485	0.4	13.3	15.4	15.3	17.3	21.5	4.0	
GINI/LISA South	3306	10.6	12.8	13.3	13.4	14.0	18.3	1.2	
GINI/LISA North	2491	15.8	16.9	17.3	17.4	17.8	21.5	0.9	
LISA East	688	0.4	17.2	17.8	17.5	18.34	20.1	1.2	
PM2.5 absorbance (10−5/m)									
Total population	5797	1.0	1.2	1.5	1.5	1.7	3.6	0.5	
GINI/LISA South	3306	1.3	1.6	1.7	1.7	1.8	3.6	0.2	
GINI/LISA North	2491	1.0	1.1	1.2	1.2	1.3	3.1	0.2	
LISA East	-	-	-	-	-	-	-	-	
Ozone (µg/m3)									
Total population	6604	32.3	39.6	42.9	42.5	44.8	59.4	5.2	
GINI/LISA South	3362	34.1	44.1	44.7	45.8	45.5	59.4	1.4	
GINI/LISA North	2551	32.3	34.4	38.0	38.2	41.4	54.3	7.0	
LISA East	691	38.0	41.0	41.8	41.9	42.5	52.9	1.5	

Total and area-specific associations

Crude and adjusted associations between air pollution concentrations at the birth address and the prevalence of health outcomes were similar (adjusted associations provided in Table 4). The area-specific results were heterogeneous. In GINI/LISA North, the estimates for allergic rhinitis and eye and nose symptom prevalence were elevated for PM2.5 mass (1.22 [0.99, 1.50] and 1.19 [0.99, 1.43], respectively). In LISA East, the estimates for ozone were elevated for all four outcomes, and that for allergic rhinitis and nose and eye symptom prevalence reached statistical significance (1.30 [1.02, 1.64] and 1.35 [1.16, 1.59], respectively). For GINI/LISA South, two associations with aeroallergen sensitization were significant (0.84 [0.73, 0.97] for NO2 and 0.87 [0.78, 0.97] for PM2.5 absorbance), as well as the association between allergic rhinitis and PM2.5 absorbance (0.83 [0.72, 0.96]). Given the heterogeneous area-specific effects, and the large influence of the GINI/LISA South cohort which represents 50.9% of the study population, risk estimates for the total population were generally null.

Table 4 Total and area-specific associations between air pollutants estimated to the birth address and health outcomes during the first 10 years of life.*

	Total population	GINI/LISA South	GINI/LISA North	LISA East	
	N	OR [95% CI]	N	OR [95% CI]	N	OR [95% CI]	N	OR [95% CI]	
NO2									
Asthma	4585	0.89 [0.73, 1.08]	2524	0.86 [0.62, 1.18]	1545	0.95 [0.77, 1.18]	516	1.02 [0.69, 1.50]	
Allergic rhinitis	4586	0.96 [0.85, 1.09]	2525	0.86 [0.71, 1.02]	1545	1.10 [0.96, 1.26]	516	1.18 [0.91, 1.54]	
Eyes and nose symptoms	4586	0.96 [0.87, 1.05]	2525	0.90 [0.78, 1.04]	1545	1.03 [0.92, 1.15]	516	0.96 [0.75, 1.24]	
Aeroallergen sensitization	3013	0.92 [0.84, 1.01]	1689	0.84 [0.73, 0.97]	975	1.06 [0.93, 1.21]	349	1.12 [0.86, 1.44]	
PM 2.5 mass									
Asthma	4585	0.97 [0.59, 1.58]	2524	0.96 [0.74, 1.25]	1545	0.89 [ 0.64, 1.23]	516	1.06 [0.83, 1.35]	
Allergic rhinitis	4586	0.87 [0.60, 1.26]	2525	0.89 [0.76, 1.06]	1545	1.22 [0.99, 1.50]	516	0.95 [0.81, 1.11]	
Eyes and nose symptoms	4586	0.93 [0.67, 1.29]	2525	0.96 [0.84, 1.10]	1545	1.19 [0.99, 1.43]	516	0.93 [0.83, 1.05]	
Aeroallergen sensitization	3013	1.10 [0.83, 1.45]	1689	1.01 [0.89, 1.14]	975	1.14 [0.95, 1.37]	349	1.03 [0.89, 1.19]	
PM 2.5 absorbance									
Asthma	4069	0.82 [0.55, 1.21]	2524	0.94 [0.75, 1.18]	1545	0.88 [0.67, 1.14]	-	-	
Allergic rhinitis	4070	0.75 [0.58, 0.96]	2525	0.83 [0.72, 0.96]	1545	1.00 [0.84, 1.20]	-	-	
Eyes and nose symptoms	4070	0.91 [0.75, 1.10]	2525	0.93 [0.84, 1.04]	1545	1.02 [0.88, 1.19]	-	-	
Aeroallergen sensitization	2664	0.82 [0.68, 0.99]	1689	0.87 [0.78, 0.97]	975	1.02 [0.89, 1.18]	-	-	
Ozone									
Asthma	4649	1.20 [0.98, 1.48]	2569	1.05 [0.98, 1.13]	1562	1.23 [0.78, 1.94]	518	1.10 [0.86, 1.42]	
Allergic rhinitis	4650	1.02 [0.90, 1.16]	2570	1.00 [0.96, 1.05]	1562	0.92 [0.68, 1.27]	518	1.30 [1.02, 1.64]	
Eyes and nose symptoms	4650	0.97 [0.87, 1.08]	2570	1.00 [0.96, 1.03]	1562	0.77 [0.57, 1.04]	518	1.35 [1.16, 1.59]	
Aeroallergen sensitization	3049	0.99 [0.89, 1.09]	1716	1.01 [0.98, 1.04]	982	0.79 [0.61, 1.02]	351	1.17 [0.96, 1.43]	
Notes.

Bold = statistically significant result (p-value < 0.05).

* Odds ratios are calculated per interquartile increase of each air pollutant. Models are adjusted for sex, age, parental history of atopy, parental education, older siblings, maternal smoking during pregnancy, smoke exposure in the home, contact with furry pets, use of gas stove for cooking, home dampness or indoor mold, intervention participation, cohort and area (total models only).

Sensitivity analyses

Associations with sensitization did not differ when aeroallergens were stratified into indoor and outdoor categories. Analyses which considered atopic asthma and more general allergic rhinitis (doctor diagnosis or nose and eye symptoms) as alternate outcomes yielded similar associations. The results remained robust when the models were further adjusted for the percent total green space and population density in a 500 m buffer around the home address (not done for LISA East) and upon adjustment for pneumonia infections in the first two years of life. Additional analyses stratified by sex, parental history of atopic disease or smoke exposure in the home during early-life did not reveal a vulnerable subgroup.

Air pollution concentrations estimated to the birth, six and 10 year addresses were generally highly correlated with one another. Consequently, the risk estimates obtained using air pollution concentrations estimated at the six and 10 year home addresses, and using the average of the birth, six and 10 year air pollution estimates, were similar to those reported for air pollutants estimated at the birth address (Fig. S2 to Fig. S4). When the pooled analyses were run stratified by moving status (never moved versus moved at least once), the risk estimates did not differ substantially between groups.

As air pollution concentrations were also generally highly correlated across time, we could only examine the potential relative importance of varying time periods of exposure, using models including both birth and six or 10 year address pollution concentrations, for a few time-point combinations. The Pearson correlation coefficient was <0.70 for two time-point combinations in the total data (between NO2 assessed at the birth and six year addresses as well as between NO2 assessed at the birth and 10 year addresses), and seven, three and three time-point combinations in the GINI/LISA South, GINI/LISA North and LISA East area-specific datasets. It was not possible to decipher a consistent trend as to which time period may be most important from the results of these models.

Discussion

In a longitudinal analysis of two German birth cohorts followed for 10 years, we did not find consistent evidence that TRAP exposure increases the risk of asthma, allergic rhinitis or aeroallergen sensitization in later childhood. The risk estimates for children living in two of the areas investigated (GINI/LISA North and LISA East) were null or elevated and those for the third (GINI/LISA South) tended to be below one. Given the heterogeneous area-specific effects, the risk estimates for the total population were inconclusive.

The factors driving the differing associations observed across areas are unknown. The sources of pollutants which likely differ by area may be one explanation; air pollution in all three areas is predominantly attributable to traffic-sources but industry also contributes to air pollution levels in GINI/LISA North (Beelen et al., 2013; Eeftens et al., 2012a). It is also possible that residual confounding may be influencing the results. We attempted to adjust for individual-level socio-economic status using parental education as a proxy in the final models, and marital status at the time of birth and household income per person (calculated according to Sausenthaler et al., 2011) in sensitivity analyses, but these indicators may be imperfect markers of socio-economic factors. Socio-economic data at the neighborhood level is not available for a large proportion of participants, but has been shown not to strongly influence associations between air pollution and allergic outcomes in recent cross-sectional multi-center European studies (A Mölter, A Simpson, D Berdel, B Brunekreef, A Custovic, J Cyrys, J de Jongste, F de Vocht, E Fuertes , U Gehring, O Gruzieva, J Heinrich, G Hoek, B Hoffmann, C Klümper, M Korek, T Kulbusch, S Lindley, D Postma, C Tischer, A Wijga, G Pershagen, R Agius, unpublished data, and Gruzieva et al., in press). Effect estimates also remained robust upon adjustment for surrounding green space and population density.

The overall null findings reported here for the total population are in line with those of two large recent multi-center European studies on asthma (A Mölter, A Simpson, D Berdel, B Brunekreef, A Custovic, J Cyrys, J de Jongste, F de Vocht, E Fuertes , U Gehring, O Gruzieva, J Heinrich, G Hoek, B Hoffmann, C Klümper, M Korek, T Kulbusch, S Lindley, D Postma, C Tischer, A Wijga, G Pershagen, R Agius, unpublished data) and sensitization (Gruzieva et al., in press), in which GINI/LISA South and GINI/LISA North were included. The current work differs from these previous studies in several respects. The current study uses a longitudinal analytical approach to optimize the use of the long-term prospectively collected health outcome data from three to 10 years, whereas the previous two studies examined cross-sectional associations at two time points. Furthermore, the two multi-center studies did not include the LISA East area, nor were allergic rhinitis or nose and eye symptoms considered as outcomes or ozone as an air pollutant. With respect to studies with a long-term longitudinal design, our findings for the total study population are in contrast to the positive associations observed between TRAP and asthma, as well as allergic rhinitis among non-movers, in a Dutch birth cohort followed for eight years (Gehring et al., 2010), but in line with the null associations for sensitization found in this Dutch cohort and in a similar Swedish birth cohort followed for eight years (Gruzieva et al., 2012).

The area-specific findings reported for the GINI/LISA North area are generally in line with a previous study in this area which examined associations with outcomes up to six years of age (Krämer et al., 2009). For GINI/LISA South, the current findings are more challenging to reconcile with those reported for outcomes up to six years of age (Morgenstern et al., 2008). The sample sizes differ slightly between this previous and the current analyses (3577 and 3941 children included, respectively), the definitions were not identical for all outcomes and there were some differences in the methodologies of the exposure assessment. In the previous work, exposure estimates were derived using a LUR model developed as part of the TRAPCA project (Brauer et al., 2003). This initial model was subsequently applied to the GINI/LISA South metropolitan area (TRAPCA II), which includes the city of Munich and the surrounding districts (Morgenstern et al., 2007). Exposure estimates derived from this TRAPCA II model were significantly positively associated with several health outcomes at six years among GINIplus and LISAplus participants living in GINI/LISA South (Morgenstern et al., 2008). In contrast, the current analysis for this area, which yielded null or negative associations, is based on estimates derived from a different LUR model developed almost a decade later as part of the ESCAPE collaboration. The ESCAPE models explain more variation than the TRAPCA II models and the root-mean standard errors of the ESCAPE models are lower than for TRAPCA II. The distribution of PM2.5 mass and absorbance concentrations are similar between the two datasets, however, the NO2 estimates are lower in the more recently derived ESCAPE dataset. The lower estimated NO2 concentrations may reflect true decreases as the air pollution measurements for the ESCAPE models were taken a decade after those for the TRAPCA models and actual air pollution levels have decreased in GINI/LISA South during this time.

To date, very few studies have reported on associations with ozone. In the current study, results were generally non-significant, with only two positive associations found for LISA East. A few factors may have hindered our ability to detect consistent associations. First, the resolution of the database (1 × 1 km) used to estimate ozone concentrations to the home address was lower than for the other pollutants. Second, as the spatial distribution of ozone is more even than for other pollutants, with the exception of areas very close to traffic, detecting true associations may be more difficult. Third, ozone concentrations are likely higher in rural areas compared to areas with greater traffic densities. However, effect estimates remained similar when the analyses were stratified into the inner city of Munich and surrounding areas (for GINI/LISA South) or when those not living in the city area were excluded (for GINI/LISA North).

Although the present work is among the very few studies which can utilize such a rich and large longitudinal dataset, certain limitations should be acknowledged. Participation bias is always a concern for cohorts with a long follow-up. Children included in this study differed from those in the initial birth cohort with regard to several characteristics and this non-random retention of participants may have affected the effect estimates. Of the 9086 children who were recruited in the GINIplus and LISAplus cohorts at birth, 5078 participated in the 10 year follow-up (55.9%). Outcome misclassification is also a concern when analyzing data collected by questionnaires, but objective measures of aeroallergen sensitization were available for 55.3% (3655/6604) of the study population. No systematic differences between the results for the three parent-reported outcomes and the objective aeroallergen sensitization outcome are apparent. Furthermore, a positive response for asthma and allergic rhinitis was based on a parental-report of a doctor diagnosis and not only on a report of symptoms. The responses at the older ages are also likely more accurate as allergic disorders are easier to diagnose in later childhood. Finally, the data used to inform the LUR regression models for GINI/LISA South and North were collected approximately a decade after the commencement of the birth cohorts (∼ five years for LISA East) under the implicit assumption that the spatial variability in air pollution estimates would not have changed since the baseline periods of the cohorts. Three studies provide evidence supporting this assumption for NO2 over a period of seven to twelve years (Wang et al., 2013; Cesaroni et al., 2012; Eeftens et al., 2011).

A greater importance of early-life exposures has been hypothesized as a possible explanation for why positive associations with TRAP and sensitization have been found at four years of age but not at eight years in a Swedish birth cohort (Gruzieva et al., 2012), a finding that was also observed in a Dutch birth cohort (Gehring et al., 2010). One previous study has considered the relative importance of TRAP exposure timing and reported that NO and PM10in utero exposures have an independent effect from post-birth exposures, although it was not possible for the authors to conclude which period may be most important (Clark et al., 2010). Similarly, the high correlation between exposures at birth and those at six and 10 years in the current study rendered it challenging to disentangle the effects of these distinct exposure periods.

Although we conducted several sensitivity analyses to reduce potential moving-related exposure misclassification and noted no changes in the results, it is possible that some exposure misclassification remains, especially at the older ages when children spend a larger proportion of their time at school. However, a Swedish and French study showed that exposures from traffic assessed at the home address are good approximations of those at schools, possibly because schools tend to be located in the close vicinity of homes (Gruzieva et al., 2012; Reungoat et al., 2005). Additionally, a study conducted in the United States found little differences between time-weighted averages of diesel exposures estimated at all addresses where a child spent more than eight hours per week and those estimated only at the home address (Ryan et al., 2008).

Conclusions

We did not find consistent evidence that TRAP increases the risk of childhood asthma or allergic diseases in later childhood using data from German birth cohort participants followed for 10 years. Heterogeneous results were noted across the three geographical areas investigated.

Supplemental Information

Figure S1 Flow chart of study population

Click here for additional data file.

Figure S2 Total and area-specific associations between NO2 (A), PM2.5 mass (B), PM2.5 absorbance (C) and ozone (D) estimated at the age six-year address with doctor diagnosed asthma (red squares), doctor diagnosed allergic rhinitis (purple triangles), eye and nose symptoms (blue stars) and aeroallergen sensitization (black triangles). OR’s are calculated per interquartile increase of each air pollutant. Models are adjusted for sex, age, parental history of atopy, parental education, older siblings, maternal smoking during pregnancy, smoke exposure in the home, contact with furry pets, use of gas stove for cooking, home dampness or indoor mold, intervention participation, cohort and area (total models only).

Click here for additional data file.

Figure S3 Total and area-specific associations between NO2 (A), PM2.5 mass (B) and PM2.5 absorbance (C) estimated at the age 10-year address with doctor diagnosed asthma (red squares), doctor diagnosed allergic rhinitis (purple triangles), eye and nose symptoms (blue stars) and aeroallergen sensitization (black triangles). OR’s are calculated per interquartile increase of each air pollutant. Models are adjusted for sex, age, parental history of atopy, parental education, older siblings, maternal smoking during pregnancy, smoke exposure in the home, contact with furry pets, use of gas stove for cooking, home dampness or indoor mold, intervention participation, cohort and area (total models only).

Click here for additional data file.

Figure S4 Total and area-specific associations between NO2 (A), PM2.5 mass (B) and PM2.5 absorbance (C) averaged at the birth, six and 10 year addresses, and ozone (D) averaged at the birth and six year addresses with doctor diagnosed asthma (red squares), doctor diagnosed allergic rhinitis (purple triangles), eye and nose symptoms (blue stars) and aeroallergen sensitization (black triangles). OR’s are calculated per interquartile increase of each air pollutant. Models are adjusted for sex, age, parental history of atopy, parental education, older siblings, maternal smoking during pregnancy, smoke exposure in the home, contact with furry pets, use of gas stove for cooking, home dampness or indoor mold, intervention participation, cohort and area (total models only).

Click here for additional data file.

Table S1 Characteristics of study participants with available serology data (N = 3655).

Click here for additional data file.

GINIplus study group

Helmholtz Zentrum München, German Research Center for Environmental Health, Institute of Epidemiology, GINI/LISA South (Heinrich J, Wichmann HE, Sausenthaler S, Zutavern A, Chen CM, Schnappinger M, Rzehak P); Department of Pediatrics, Marien-Hospital, GINI/LISA North (Berdel D, von Berg A, Beckmann C, Groß I); Department of Pediatrics, Ludwig-Maximilians-University, GINI/LISA South (Koletzko S, Reinhardt D, Krauss-Etschmann S); Department of Pediatrics, Technical University, GINI/LISA South (Bauer CP, Brockow I, Grübl A, Hoffmann U); IUF-Institut für Umweltmedizinische Forschung at the Heinrich-Heine-University, Düsseldorf (Krämer U, Link E, Klümper C); Centre for Allergy and Environment, Technical University, GINI/LISA South (Behrendt H).

LISAplus study group

Helmholtz Zentrum München, German Research Center for Environmental Health, Institute of Epidemiology, GINI/LISA South (Heinrich J, Wichmann HE, Sausenthaler S, Chen CM, Schnappinger M); Department of Pediatrics, Municipal Hospital ‘St. Georg’, LISA East (Borte M, Diez U), Marien-Hospital GINI/LISA North, Department of Pediatrics, GINI/LISA North (von Berg A, Beckmann C, Groß I); Pediatric Practice, Bad Honnef (Schaaf B); Helmholtz Centre for Environmental Research-UFZ, Department of Environmental Immunology/Core Facility Studies, LISA East (Lehmann I, Bauer M, Gräbsch C, Röder S, Schilde M); University of LISA East, Institute of Hygiene and Environmental Medicine, LISA East (Herbarth O, Dick C, Magnus J); IUF-Institut für Umweltmedizinische Forschung, Düsseldorf (Krämer U, Link E, Klümper C); Technical University GINI/LISA South, Department of Pediatrics, GINI/LISA South (Bauer CP, Hoffmann U); ZAUM-Center for Allergy and Environment, Technical University, GINI/LISA South (Behrendt H, Grosch J, Martin F).

Additional Information and Declarations

Competing Interests

Author Contributions

Human Ethics

Joachim Heinrich is an Academic Editor for PeerJ.

Elaine Fuertes conceived and designed the experiments, performed the experiments, analyzed the data, wrote the paper.

Marie Standl analyzed the data, contributed reagents/materials/analysis tools.

Josef Cyrys, Dietrich Berdel, Andrea von Berg, Carl-Peter Bauer, Ursula Krämer, Irina Lehmann and Sibylle Koletzko contributed reagents/materials/analysis tools.

Dorothea Sugiri conducted the GIS work (address assignments).

Chris Carlsten supervised work and helped with data/results interpretation and with statistical design.

Michael Brauer conceived and designed the experiments.

Joachim Heinrich conceived and designed the experiments, contributed reagents/materials/analysis tools.

The following information was supplied relating to ethical approvals (i.e., approving body and any reference numbers):

For the GINI south study area: Bavarian Board of Physicians (Reference # for 10 year follow-up: 01212)

For the GINI north study area: Board of Physicians of North-Rhine-Westphalia (Reference # for 10 year follow-up: 2003355)

For the LISA south study area: Bavarian Board of Physicians (Reference # for 10 year follow-up: 07098)

For the LISA north study area: Board of Physicians of North-Rhine-Westphalia (Reference # for 10 year follow-up: 2008153)

For the LISA east study area: University of Leipzig (Reference # for 10 year follow-up: 345/2007).

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
