# Peer review of "A longitudinal analysis of associations between traffic-related air pollution with asthma, allergies and sensitization in the GINIplus and LISAplus birth cohorts"

_PeerJ, doi:10.7717/peerj.193_

## Round 0.1 · original submission · Minor Revisions

Sorry for the delay but it was difficult to find a reviewer

·

Basic reporting

No comments.

Experimental design

This manuscript reports on the relationship between exposures to traffic related air pollutants (TRAP) and asthma and allergic outcomes in Germany. It is clearly written and the methods are well described, with the exception that the statistical analyses could use more detail for the cross sectional analyses if they remain in the paper (see next section). The paper is laudable in that the authors have harmonized data across several birth cohorts all recruited around the same years in order to increase sample size and geographic variability. The sample sizes, while large, still result in low numbers of diagnoses in LISA East, and asthma in all locations. Nevertheless the confidence limits are relatively precise. This paper is also important because of its use of sophisticated pollutant models that reveal, bottom line, relatively conclusive negative results.

Validity of the findings

Potential confounders should be included in a model because they are related to both the exposure and the outcome. Therefore it is unclear why some of the variables were included, such as sex of the child or presence of an older sibling. Are variables such as these, and others such as gas stove use and parental history of atopy, related to outdoor pollutant exposure in any way? Did the authors test to see whether the addition of a confounder materially changed the risk estimates for the pollutants and outcomes, such that it warranted keeping the variable in the model?

The authors could consider the possibility of effect modification. Could the relationship between pollutants and outcomes be different for subgroups such as males, or those with a positive family history or ETS exposure?

More information is needed for the non-German reader about the characteristics of the geographic areas of the cohorts. It appears from the cohort characteristics that the North area is of lower socioeconomic status, the South, the highest and East somewhere in between. Are socioeconomic data less crude than one parent with more than 10 years of education (yes/no) needed, or available? Presumably socioeconomic status is highly related to pollutant level and perhaps the outcomes? Would there be any differences in medical care use and therefore probability of obtaining a doctor diagnosis, between the areas? What are the pollutant sources in the areas?

Most importantly, what has happened to pollution levels over time in the three areas, and does this differ by area? The Escape models used for North and South are based on data collected 10-13 years after the births occurred. The model used for the East population is based on data collected just a few years after the births, so even though the model is noted to have limitations that the Escape models do not, the measurements are much closer in time to the births. Non-systematic exposure misclassification, which may be especially a problem in the North and South area if pollution has decreased over time (which is indicated in the Discussion for the South, lines 341-342), would tend to drive associations towards the null, and most of the associations are consistent with 1.0. These are critical limitations that need to be included in the Discussion.

Results, page 9, line 238. These results for the total population are completely driven by GINI/LISA South, so I don’t think they are worth mentioning except perhaps in that context. The results mentioned for GINI/LISA North (also in Discussion lines 321-322) are also not really notable. They are not statistically significant even though it is easy to obtain significance with the large sample and high prevalence of allergic rhinitis and eye/nose symptoms. Given the large number of comparisons, the large sample sizes for most estimates, the variation across geographies for the same associations, and the relatively weak estimates with confidence limits that include or are very close to 1.0, in this reviewer’s view, Table 4 is basically a table of null results.

The paper is focused on exposure to pollutants at birth and outcomes at age 10, but in Figure 2 suddenly switches to cross-sectional analyses of pollutants estimated by addresses at ages between 3 and 10 and the same outcomes. These analyses are less clear. If a child was diagnosed at age 4 with asthma and then moves, is he included in the asthma prevalence estimates at age 10 with the pollution measures at his new address? I see no reason to include cross-sectional analyses when longitudinal data are available.

Figure 1: These graphs appear to be cumulative incidence data (ever history of a doctor diagnosis), so it is unclear to me why the prevalence of asthma (and once for allergic rhinitis) goes down in some years.

Assuming most of the children with asthma were also atopic, it would be useful to repeat the analyses for the specific phenotype of atopic asthmatics to reduce misclassification, presuming this could not be done due to sample size for non-atopic asthmatics.

Are the data such that analyses can be repeated for the combined category of doctor diagnosed and undiagnosed hay fever since many children do not receive a diagnosis for these conditions (or is that not the case in Germany)?


Minor Comments:

Intro, second paragraph. I think validity of exposure is just as difficult, if not more so, than accuracy of outcome measurement.

Since the authors couldn’t evaluate exposure timing due to the high correlation across time points in most cases, and couldn’t decipher a trend, they may just want to mention this in the Discussion (and maybe the Methods) but not in the Introduction.

Discussion, line 368, except for the cross-sectional analyses (see my concern above), they were not really able to use the pollutant data at three time points successfully.

Discussion: line 360: I am not sure what is meant by the concern of recall bias. Individuals are not providing pollution data, only home addresses, which should be easily collected via bi-annual questionnaires and not vary in validity by those with positive outcomes versus no outcome.

I am not sure that Table 1 needs to include source data that is not used in the paper.

Additional comments

No comments

·

Basic reporting

No comments.

Experimental design

Line 122: "For the seven to 10 year follow-ups of GINIplus and for the entire LISAplus follow-up, this information ..." As there is no follow-up between six and 10 years, it seems like this part should be left out: "the seven to"

Line 138: "A test was defined as positive if the immunoglobulin E value was greater or equal to 0.35 kU/L." This definition needs a reference to document its validity.

Line 141: The size of the subpopulation with sensitization tests should be added, separately for 6 and 10 years of age.

Line 174: The definition of parental education is rather coarse. Is it possible to apply a three category definition instead? Furthermore, socioeconomic factors (SES) are usually related to environmental variables such as air pollution. Thus, individual SES should be properly adjusted for when studying health effects of air pollution. Is information about parental marital status, household income or ethnic origin available?

Validity of the findings

Line 293: The authors discuss whether residual confounding may be influencing the results, although several individual level confounders were considered. This discussion is appropriate. I suggest to mention individual SES especially (see my comment above). Generally, individual SES variables are more important than SES variables at neighborhood level.

Line 359: “At the 10 year follow-up, the retention rate was 55.9% for both cohorts combined.” How is 55.9% achieved?

Line 360: “… , but objective measures of aeroallergen sensitization were available for 55.3% of the study population.” How is 55.3% achieved? (n/N) should be added.

Additional comments

The study is a nicely designed and very well written paper about the relationship between air pollution and allergic outcomes in a birth cohort followed up to 10 years. Publication is recommended.

---

## Round 0.2 · accepted · Accept

excellent paper. Thank you for submitting the paper to PeerJ